# Decolonising humanitarian health: A scoping review of practical guidance

**Amber Clarke**[1]◉*, **Katharina Richter**[2]◉, **Michelle Lokot**[3‡], **Althea-Maria Rivas**[4],
**Sali Hafez**[3], **Neha S. Singh**[3‡]

**1** FAIR Network, London School of Hygiene and Tropical Medicine, London, United Kingdom, **2** School of Sociology, Politics and International Studies, University of Bristol, Bristol, United Kingdom, **3** Health in Humanitarian Crises Centre, London School of Hygiene and Tropical Medicine, London, United Kingdom, **4** SOAS University of London, London, United Kingdom

◉ These authors contributed equally to this work.
‡ These authors also contributed equally to this work.
* amber.clarke1@lshtm.ac.uk

## Abstract

Despite growing calls and efforts to decolonise global and humanitarian health, there is limited practical guidance for researchers, educators and practitioners on how to do so. This paper fills this gap by offering a narrative exploration of key recommendations on decolonising global/humanitarian health research, partnerships, teaching, organisational structures and other practices. We present concrete guidelines to support humanitarian actors in decolonising their work. We used a scoping review method. The search strategy was built on three overarching themes: decolonising, global health/health and humanitarian crises. We combined a MEDLINE and Web of Science database search with a grey literature search. In total, we screened abstracts and titles of 533 documents, excluding records that did not specifically refer to 'decolonising,' humanitarian and/or global health. We assessed full texts of 58 documents for eligibility, excluding documents that did not include practical recommendations. In total, 15 documents were included in this review. We identified five key themes: organisational structure, strategy and engagement; research partnerships and conceptualisations; funding for research and projects; the research lifecycle; and teaching and the curriculum. The principal finding is that humanitarian actors can decolonise their work by decentralising power, redistributing resources, critically reflecting on their work in the context of the broader socio-political landscape and recovering, centring and valuing marginalised Global South perspectives. Race was not a central analytical category in the reviewed literature, despite being an integral part of historical background narratives. Future research should reflect on practical steps towards racial justice in global/humanitarian health and be focused on ensuring that efforts towards "localisation" or "equitable partnerships" in global health are linked to decolonisation efforts, including in humanitarian health research. Our review underscores the importance of drawing on knowledge created by and for actors based in the Global South.

**Data Availability Statement:** All data are in the manuscript and supporting information files.

**Funding:** This work was supported by the LSHTM/Wellcome Institutional Strategic Support Fund (Wellcome grant reference 204928/Z/16/Z) (NS) and funds provided by the LSHTM Executive Office through the FAIR x LSHTM commissioning process (AC). The funders had no role in study design, data collection and analysis, decision to publish, or preparation of the manuscript.

**Competing interests:** The authors have declared that no competing interests exist.

## Introduction

The central argument for continuing efforts to decolonise structures of domination is that the 19[th] and 20[th] century political decolonisation processes ought to be considered incomplete. Gaining political sovereignty did not necessarily address continuing social, cultural and economic hierarchies in former colonies [1,2]. This coloniality, that is, the logic of domination, is argued to be exercised today via the interaction of knowledge, racism, patriarchy and capital [3,4]. Valid knowledge production is often confined to the techno-scientific realm, and precludes the acknowledgement of indigenous, peasant, or Afro-descendant knowledge systems [5]. Coloniality thereby continues to shape social and cultural, as well as political and economic systems across the globe [6], including the fields of international development and humanitarian assistance.

Development and humanitarian assistance–as forms of aid–have their roots in colonial ideologies around the need to bring Western-based ideas of progress, scientific advances and industrial progress to the Global South [7]. Development has been associated with long-term approaches that are often grounded in addressing the root causes of poverty and inequality [8]. Humanitarian assistance, in contrast, involves a more short-term emergency response to natural disasters, war, famine and other humanitarian catastrophes based on humanitarian principles of humanity, impartiality, neutrality and independence [9]. However, the distinction between these two types of aid is blurry, more so with the emergence of what has been called the "humanitarian-development nexus" [10]. Development and humanitarian assistance are often linked and may perpetuate colonial narratives about non-white populations requiring rescue or being underdeveloped because they lacked the technology and knowledge systems of the "developed" world [11].

Colonial-era, racialised social classifications of the world's population still persist, resulting in racism and racialising discourses influencing research, policy, and practice [12–17]. What Pailey calls the "White Gaze of Development" is built on a system of white supremacy that shapes aid more broadly [18]. Whiteness signifies expertise and facilitates access to structural power and privilege. For example, non-white experts and local staff are often employed precariously and are marginalised in decision-making compared to their white counterparts [18–20]. Expertise and political, economic and socio-cultural processes are often measured against a standard that is white, Western, and often male [18]. Race therefore remains a foundational category to aid, pervading its reason for being, institutional structures, and every-day practices [12]. Within the humanitarian sector specifically, the concept of localisation [which broadly refers to shifting power and funding to national actors] became a mainstream reform issue at the 2016 World Humanitarian Summit [21,22]. Despite commitments to shift power by 67 signatories from 25 Member States, 26 NGOs, 12 UN agencies, two Red Cross/Red Crescent movements, and two inter-governmental organisations, progress has been limited, reaffirming calls to go further than localisation and decolonise aid [21,23–27].

Within development and humanitarian assistance, decolonising has sometimes been conflated with other terms such as ensuring "equity, diversity and inclusion", "equitable partnerships" or "localisation". [22] while these approaches seek to tackle specific inequities within the humanitarian system, they are concerned with specific symptoms of coloniality–such as unequal access to funding, inequitable and unethical authorship arrangements or asymmetrical partnerships–without addressing the root causes of these manifestations. There is increased recognition that addressing modern-day legacies of colonialism, empire, racism and patriarchy first requires recognising and making visible the deeply problematic assumptions and attitudes that often underpin development and humanitarian practice [28]. Debates persist about whether aid should be abolished entirely or if it can be reformed to tackle these power

hierarchies [27]. These debates are mirrored in efforts to decolonise Higher Education [29–31]. Campaigns to acknowledge and work towards the atonement of some of the academy's colonial legacies have been led by students and social movements across the world, from Rhodes Must Fall in South Africa and the University of Oxford to the global reckoning of race relations spurred on by the Black Lives Matter movement [32].

There remains a lack of clarity on what decolonising development and humanitarian assistance might mean in practice. In this paper, we understand decolonising practices broadly as efforts that highlight and decouple humanitarian practices from colonial power relations and logics [33]. These efforts, for example, might seek to expose the geopolitical locus of knowledge production and give a platform to people and forms of knowledge that have been racialised and marginalised [34,35]. Throughout this project, we make use of the active verb "decolonising", as opposed to the static noun "decolonisation", to indicate an inherently incomplete, ongoing and evolving process, as opposed to an end that can be achieved at a fixed point in time. At the same time, we argue that this ongoing and evolving process must also be inherently practical and achievable.

Despite growing calls and efforts to decolonise humanitarian and global health, at the time of writing, practical guidance on how to do so had not been reviewed. Some recommendations on how to decolonise humanitarian and global health have recently been published and are available in various accessible formats [26]. While this also includes a scoping review of decolonising the field of global health [36], these documents and calls have not been distilled into practical and workable guidelines for researchers, educators and practitioners aiming to decolonise their work in the field of humanitarian health. The present review therefore fills a gap in the literature on decolonising humanitarian health, specifically the limited practical guidance on how to decolonise research, teaching, organisational structure and other practices. The review is therefore relevant for those researching, teaching and working in the field of humanitarian health.

## Methods

This scoping review was conducted as part of a collaborative project between The Fight Against Institutional Racism (FAIR) network and the Health in Humanitarian Crises Centre (HHCC) at the London School of Tropical Hygiene and Medicine (LSHTM), where the majority of authors are writing from. LSHTM was established in 1899 with a direct view to aid and abet British colonialism [37]. The FAIR Network at LSHTM is a legacy of the 2020 Black Lives Matter protests and efforts by staff and students to address racism and coloniality within LSHTM. The FAIR Network is an independent network committed to facilitating dialogue and action around issues of racism and colonial legacies in global public health. Two of its members have led this review. The review presents a narrative exploration of the key recommendations and guidelines on decolonising global/humanitarian health research, partnerships, teaching, organisational structures and other practices.

This review aims to analyse guidelines that support humanitarian actors in their attempts to decolonise teaching, research and practice. In line with Pailey's [18] call, we understand that race should be a central analytical category in this review and thus will extract it from the literature where possible and highlight its absence where this is the case. The research questions for this scoping review were: "What should decolonising global/humanitarian health research, teaching, partnerships, organisational structures and other practices look like in practice?"; and "What practical guidance and recommendations is available to aid researchers, educators and practitioners looking to decolonise their work in a global health/humanitarian setting?". It

aims to identify the main sources and types of evidence available for decolonising global/humanitarian health, including key recommendations and existing guidelines.

This scoping review synthesises peer-reviewed and grey literature published on key recommendations and practical guidelines on decolonising global/humanitarian health research, partnerships, teaching, organisational structures and other practices, in line with Arksey and O'Malley's [38] scoping methodology. A scoping methodology was deemed an appropriate method since the topic of practical guidelines for decolonising health "is complex or has not been reviewed comprehensively before" [39]. The scoping review was conducted between August 2022 and March 2023. We furthermore adapted the PRISMA checklist extension for our methodology [40].

We reviewed both academic and grey literature, and narrowly focused on practical guidance, charters, tools, accountability frameworks, mechanisms or recommendations, as this was the focus of the broader FAIR-HHCC research collaboration. As is common with such research centres, HHCC activities include partnerships, research and teaching. Functioning as a small organisation within a larger institution, organisational structure and practices such as recruitment and staff training practices are also a part of this work. Therefore, we included practical guidance in relation to decolonising all of these areas. We included global health as well as humanitarian health in the search terms and in the eligibility criteria, because guidance on decolonising written in/for the global health context was likely deemed relevant to humanitarian health research and practice as well. Subsequently, we refer to guidance in this review in the context of "global/humanitarian health". Finally, we only focused on literature that specifically used the term "decolonising". We did not review literature articulated in terms of (arguably) related terms such as "equitable partnerships", "equality, diversity and inclusion", "localisation", "indigenisation", "Black lives Matter" or others. The implications of this choice is addressed in the Discussion.

## Search strategy

To identify relevant peer-reviewed studies, two members of the research team (KR and ML) conducted a search of academic literature for terms related to global/humanitarian health research and decolonisation on MEDLINE and Web of Science. The search was built on three key concepts that summed up the overarching themes of this research: decolonising, global health/health and humanitarian crises. Given the broad and disparate understanding of "decolonising" and "decolonisation" search terms were broad including keywords such as: antiracism, equity, epistemic justice, and imperial. The search terms for both databases can be found in Tables A and B in S1 Appendix: Database search terms. The searches were conducted in August 2022.

To identify relevant grey literature, the lead author (AC) subsequently employed multiple search strategies. These included using the search function of the LSHTM intranet to identify relevant internal documents; searching websites of key donors (e.g. ELRHA, International Development Research Centre), international non-governmental organisations (NGOs) (e.g. Médecins Sans Frontières, Save the Children, International Rescue Committee), multilateral agencies (e.g. United Nations High Commissioner for Refugees, Africa Centres for Disease Control and Prevention), relevant think tanks and centres/research groups in all income settings (e.g. American University of Beirut, Overseas Development Institute); obtaining resources familiar to HHCC Leadership. The search terms for all grey literature included: decolonising, decolonisation, anti-racism, humanitarian research, global health, and humanitarian crises. Search results of both grey and academic literature were narrowed down during the subsequent screening process.

**Table 1. Screening criteria.**

| Abstract and Title Screening Criteria | | |
|---|---|---|
| **Category** | **Include** | **Exclude** |
| **Context** | Decolonising humanitarian or decolonising global health | Other (not related to humanitarian or global health) |
| **Organisation** | LSHTM or other research centres/groups, donor organisations, international NGOs, multilateral agencies or think tanks | None |
| **Literature type** | Reviews, opinion pieces, book chapters, dissertations, Internal organisational reports, guidance documents and blog posts | Other forms of media (podcasts, lectures, videos) |
| **Date** | Any | None |
| **Language** | English | Not available in English |
| Additional Full-Text Screening Criteria | | |
| **Content** | Practical guidance, charters, tools, accountability frameworks, mechanisms or recommendations | Other (not practical) |

## Eligibility criteria and screening

Due to the large number of records generated by the broad search terms, we decided to focus specifically on documents that explicitly reference decolonising when doing the title/abstract screening and full-text review. We therefore narrowed down the eligibility screening criteria from the broad database search terms used in the MEDLINE and Web of Science searches to only screen for documents that explicitly mention decolonising the humanitarian sector or decolonising global health.

Consequently, to select relevant studies, title/abstract screening of search was conducted by one member of the research team (KR) using the criteria laid out in Table 1. Full texts of articles included by title and abstract screening were then screened by another member of the research team (AC), applying the same criteria except for the addition of a "content" criterion (see Table 1). This additional criterion allowed us to include documents with practical guidance only, in accordance with the research questions. Screening criteria were the same for title and abstract and full-text screening, except for an additional "content" criterion for full-text screening.

## Data extraction

To extract the data, the lead author (AC), together with input from all co-authors drafted an extraction table using Google Sheets. The extraction table included descriptive information for all records and emerging themes that AC identified during the preliminary reading of included records. The descriptive information included geographic location of first-author; context (global health or humanitarian aid), organisation type (higher education institution, non-governmental organisations or think tank/research centre); a content synopsis; and the practical guidance provided (e.g. through charters, frameworks, recommendations, accountability tools or mechanisms). Additional information emerged during further review of the included documents. This was grouped into areas of work addressed in the document (iteratively organised into: organisational practice, research partnerships and conceptualisation, the research lifecycle, research and project funding, and teaching and curricula) and also included author-stated potential barriers to implementing the guidance as well as author-stated facilitators, best practices and practical examples.

## Synthesis of results and thematic construction

Following data extraction, the lead author (AC) synthesised the evidence in order to be able to collate, summarise and report the results. Evidence was synthesised by descriptive

characteristics, which provides a view of where this evidence is produced, and by whom; and thematically, by area of work addressed. This information, together with practical guidance, and barriers and facilitators to decolonising was synthesised into key overarching guidance and detailed suggested actions, and grouped thematically according to area of work addressed (organisational practice, research partnerships and conceptualisation, the research lifecycle, research and project funding, and teaching and curricula). The results section will present the detailed synthesis.

## Gaze and researcher positionalities

We acknowledge the difficulties in conducting research on the topic of decolonising global/humanitarian health from the Global North, and within an institution with an overt colonial history in particular. We also note that the use of the terms Global North and Global South are themselves inaccurate and contentious. The terms are used loosely to describe former colonising and formerly colonised nations. This binary is problematic, masking the nuances of different cultures, geographies, and histories. At the same time, global (development) discourse is set up along this dichotomy [41], so we use these terms to insert ourselves into the debate meaningfully while recognising the limitations of this binary. In this paper, when providing our own views and analysis, we choose to use Global North and South rather than the World Bank's [42] income classifications of High-Income Country (HIC) and Low/Middle Income Country (LMIC) or Low-Income Country (LIC) respectively. However, when describing guidance found in the literature, we use the original terminology of the authors which is often HIC or Western and LMIC or non-Western.

The co-authors of this study represent diverse positionalities. ML, SH, AR and NSS are women of colour with a background of living and working in humanitarian and development settings, from different regions of the world, and with personal and professional experiences of the ongoing impacts of colonialism. All the authors are situated in, and write from, academic institutions in the Global North. Our multiple, intersecting positionalities require that we ask ourselves questions which include: Are we merely reproducing colonial power relations in which researchers in the Global North are the subjects, and people in areas of global/humanitarian health interventions the objects of research? To what extent are we reproducing colonial power relations in which Global North researchers define the parameters of research, receive the funding, and bolster their academic standing by publishing a peer-reviewed paper on this topic [43]? Acknowledgement of positionality is not a static exercise, but rather part of an ongoing reflection of our own complicity in and relationship to research, knowledge extraction and colonialism [44]. One of the limitations of this review is its exclusive review of English-language literature only, which itself narrows the scope of actors, narratives and ideas that informed this review. While keeping the above considerations in mind, we are also aware that overwhelmingly, headquarters and operations, academic contributions and funding bodies in the field of global/humanitarian health are either located in, or originate from, the Global North. In order to dismantle these colonial hierarchies, both logistically and mentally, we need guidance on how those currently working in this field in the Global North can do this. Our review collates and solidifies available guidance from calls to challenge these power dynamics in global/humanitarian health. We therefore hope to make a meaningful contribution to ongoing and continuously evolving efforts to decolonise global/humanitarian research, partnerships and teaching by identifying key recommendations and practical guidance from across the academic and grey literature.

## Results

### Characteristics of included documents

The scoping review included 15 documents (6 peer-reviewed and 9 grey literature) after reviewing 533 records. Of the 15 documents included after full text review, seven documents were focused on global health [45–51]. The remaining eight documents focus on humanitarian issues [23,25,26,52–55] and research [56] respectively. Fig 1 summarises screening results and Table 2 the characteristics of all included documents.

In terms of geographical spread of the institutions of first authors, fourteen documents were written by (first) authors who were currently affiliated with Global North organisations (see Table 2 for details), and only one by an author affiliated with a Global South organisation [51]. Publications from the Global North are categorised as such because either their lead author at the time of writing was affiliated with academic institutions in the Global North [45–48,50,53,56], or they were written on behalf of, and/or published by, humanitarian organisations or higher education institutions legally registered and geographically located in the Global North [23,25,26,49,52,54,55]. The paper from the Global South [51] was categorised as such because the first author was affiliated with an academic institution located in the Global South. This description only represents the location of institutions where lead authors are based but may not reflect the identities of lead authors themselves.

Eight documents were written by members of higher education institutions and/or published in peer-reviewed journals [45–48,50,51,53,56]. By type of publisher, three were from non-governmental organisations (NGOs) from the UK [23,26,55] and one was from an international NGO [49]. Three academic and grey literature documents were written by individual authors associated with consultancies and think tanks [25,52,54].

### Key guidance and recommendations for decolonising research, partnerships and teaching in global/humanitarian health

The following sections present the key guidance and recommendations outlined in these 15 documents. Themes and recommendations are grouped by area and are summarised in Table 3.

**Organisational structure, strategy and engagement.** The first area of guidance we identified considers the structures, strategies and political engagement of humanitarian and/or global health organisations. First, global/humanitarian health institutions require a clear roadmap that focuses on producing systemic changes to move from decolonising rhetoric to practice. Khan et al. [47] recommend that this roadmap should identify how global health organisations perpetuate inequity; publish a clear list of reforms required to decolonise global health practice to ensure a more proactive and coordinated effort; and develop metrics to track progress and transparently share findings via different public channels. The authors suggest anticipating resistance by gatekeepers who likely benefit from colonial practices, and counteracting resistance by forming alliances with progressive social movements that target systemic change. Finally, they suggest that linking with, and amplifying the concerns of, workplace women's or disability networks would ground this work in an intersectional perspective.

Second, global/humanitarian health institutions based in HIC need to redistribute resources to reflect the geographical focus of their work [47,49]. This could involve relocating staff, offices and other resources to create employment opportunities in countries where research and programmes are taking place. However, organisations should avoid replacing Global North headquarters with "country offices" [26]. The latter are one of the most visible and entrenched manifestations of structural racism in the humanitarian, development and



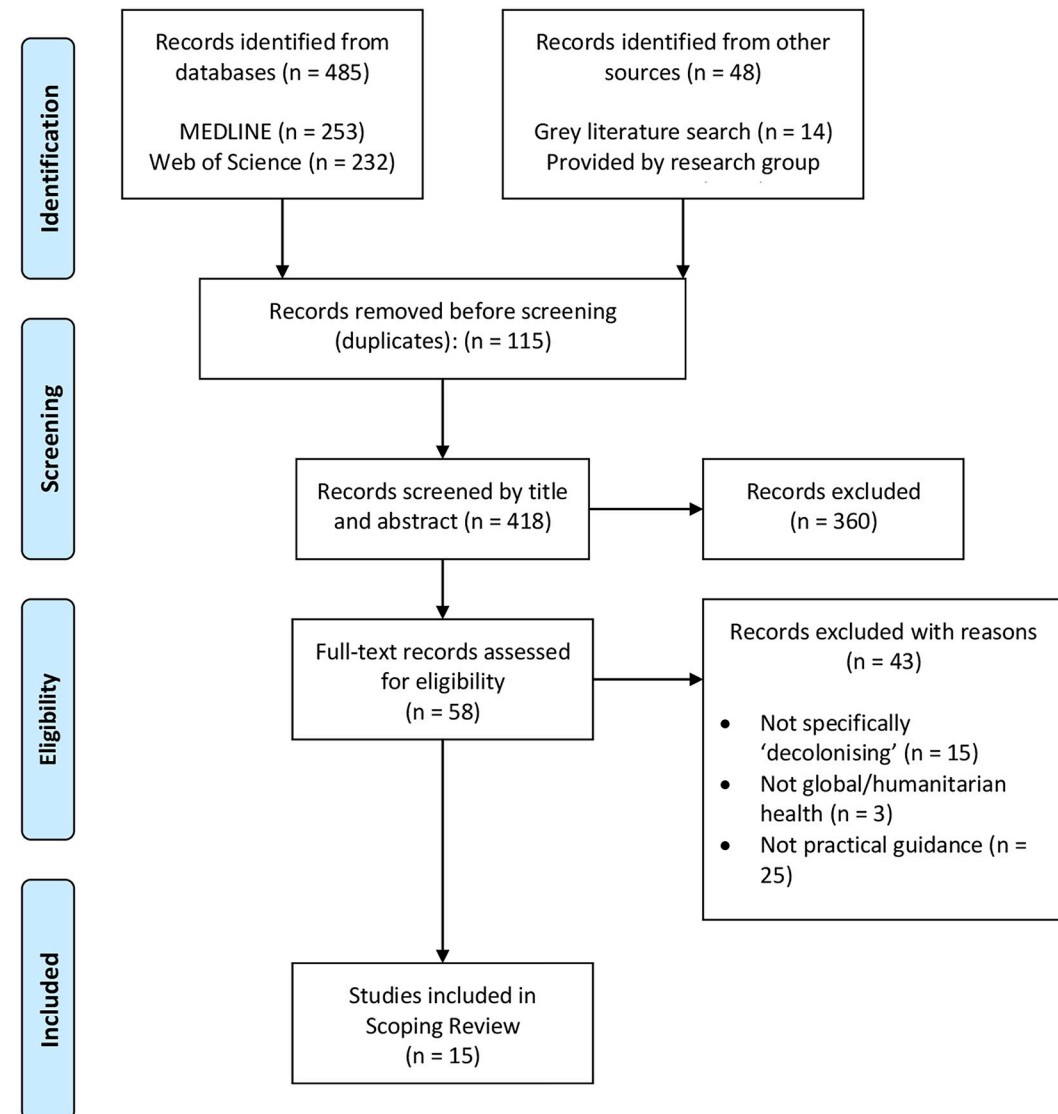

**Fig 1. PRISMA Screening Results.** PRISMA flow chart of documents included in the scoping review.

**Table 2. Characteristics of included documents.**

| Author (year) | Title | Type of article | Author location | Organisation type | Sector context | Practical area |
|---|---|---|---|---|---|---|
| Abimbola et al. (2021) [45] | Addressing power asymmetries in global health: Imperatives in the wake of the COVID-19 pandemic | Peer-reviewed journal article | Global North | Higher education institution | Global Health | General |
| Aloudat & Khan (2022) [53] | Decolonising humanitarianism or humanitarian aid? | Peer-reviewed journal article | Global North | Higher education institution | Humanitarian aid | General |
| Koum Besson (2022) [48] | How to identify epistemic injustice in global health research funding practices: a decolonial guide | Peer-reviewed journal article | Global North | Higher education institution | Global health | Research funding |
| Khan et al. (2021) [47] | Decolonising global health in 2021: a roadmap to move from rhetoric to reform | Peer-reviewed journal article | Global North | Higher education institution | Global health | General |
| Rasheed (2021) [51] | Navigating the violent process of decolonisation in global health research: a guideline | Peer-reviewed journal article | Global South | Higher education institution | Global health | Research partnerships |
| Singh et al. (2021) [56] | Research in forced displacement: guidance for a feminist and decolonial approach | Peer-reviewed journal article | Global North | Higher education institution | Humanitarian research | Research practice |
| Aid Re-imagined (2020) [52] | It's time to decolonise project management in the aid sector | Blog post | Global North | Consultancy | Humanitarian aid | Organisational practice |
| Koum Besson et al. (2022) [48] | Introduction to Decoloniality & Anti-racism in global health: Student Toolkit | Internal resource | Global North | Higher education institution | Global health | Teaching |
| Kertman (2021) [54] | Do What I Say, Not What I Do: Decolonizing Language in International Development | Blog post | Global North | Think tank/ Consultancy | International development and humanitarian aid | Organisational practice |
| Kumar (2021) [49] | Shaping a post-colonial INGO | Blog post | Global North | International NGO | Global health | Organisational practice |
| LSHTM (2022) [50] | Decolonising the Curriculum Toolkit | Internal resource | Global North | Higher education institution | Global health | Teaching |
| Narayanaswamy (2021) [23] | Decolonising Aid | Briefing | Global North | UK NGO | Humanitarian aid | Organisational practice |
| Patel (2021) [25] | Localisation, racism and decolonisation: Hollow talk or real look in the mirror? | Blog post | Global North | Consultancy | Humanitarian aid | General |
| Peace Direct (2021) [26] | Time to Decolonise Aid | Report | Global North | UK NGO | Development, humanitarian aid and peacebuilding | General |
| Start Network (2022) [55] | Anti-Racist and Decolonial Framework | Internal resource | Global North | UK based network of NGOs | Development and humanitarian aid/action | Organisational practice |

peacebuilding system, as they are almost always subordinate to Global North-headquartered organisations that are led by typically white, Western staff [26]. One example of a more equitable geographical redistribution of resources is the "networked model" recently employed by IPAS, whereby rather than having a single headquarters in the US, some IPAS offices are now independently registered [49].

Third, a key inequity-driving practice is the limited participation of LMIC experts and community representatives in governance structures and advisory bodies of organisations focusing on improving LMIC health outcomes. Shifting power to in-country and/or community experts would typically require diverse board leadership, and senior management to shift power from a centralised body to a more dispersed set of actors [47,49]. To do so, governing bodies should recruit individuals who are willing to be more critical and disrupt existing practice, as well as people from diverse gender, social, geographical and ethnic backgrounds [26,47]. These positions should be selected transparently with stakeholder input [47]. Another factor to reassess when redistributing (human) resources is the disproportionate recruitment of international staff to overseas positions [26,49]. In addition to hiring in-country staff and experts,

**Table 3. Key guidance to decolonising global/humanitarian health identified by the scoping review.**

| Area of Recommendation | Suggested actions |
|---|---|
| **Organisational structure, strategy and engagement** | |
| **Develop a roadmap outlining how to decolonise** | • To underpin a roadmap, identify and understand colonial practices and build awareness of how colonial histories have shaped organisational practices [45,47]<br>• Publish plan to decolonise and develop metrics to track progress [47]<br>• Share findings via different public channels [47] |
| **Decentralise power and redistribute resources** | • Relocate staff, offices and other resources to reflect the geographical focus of the work [47,49]<br>• Transfer resource ownership and spread operations to different offices/organisational locations across Global South, allowing researchers to work domestically [45] |
| **Reform leadership** | • Ensure the majority of leadership roles are held by people with relevant in-country (or regional) and lived experience [47]<br>• Recruit diverse individuals with critical perspectives that disrupt unequal power dynamics [25,47]<br>• Ensure input of populations affected by/working in crises settings during recruitment [26,49]<br>• Select candidates transparently [26,49]<br>• Increase representation/diversity in staff in offices in the Global North [26,45,49] |
| **Reframe communications** | • Replace passive "beneficiary" language with terms focused on reparations and social justice<br>• Update internal style guides to replace generalising, dehumanising and objectifying language [46,50,54]<br>• Mainstream race(ialisation) in communications policy similar to gender, ability and class [54]<br>• Contribute to research that can be used to challenge public perceptions of global poverty [54] |
| **Broaden allyship, solidarity and political engagement** | • Critically reflect on privilege by recognising individual intersectional identities and positionalities [45]<br>• Ensure that terms of solidarity engagement are determined by those with whom solidarity is exercised [45]<br>• Build Southern Networks, to promote Southern ownership of global health (agenda-setting, decision-making, knowledge creation etc.), while avoiding only more privileged Global South actors from accessing benefits [45]<br>• Link humanitarian aid with other social justice issues [53] |
| **Research partnerships and conceptualisation** | |
| **Prioritise regional perspectives and re-envision capacity** | • Reconsider staffing structures so that international staff are technical advisors and coordinators rather than decision-makers [47]<br>• Select International staff with adequate contextual expertise [47]<br>• Invest in capacity strengthening and sharing [52]<br>• Connect colleagues to resources and power [52] |
| **Create open, transparent, regular and formal channels of communication during research partnerships** | • Schedule regular partnership reviews and invite feedback [52]<br>• Seek consensus among project team members (data collectors included) on project strategies [51]<br>• Researchers located in the Global South are advised to insist on being part of project communications [51]<br>• Document project decisions transparently [51] |
| **Funding for Research and Projects** | |

*(Continued)*

**Table 3.** (Continued)

| Area of Recommendation | Suggested actions |
|---|---|
| **Encourage funding localisation** | • Increase the quality and quantity of funding directed to local actors [45]<br>• Include local civil society organisations in developing Calls for Proposals and reviewing applications [48] |
| **Address epistemic injustice in funding proposals** | • Redefine evaluation criteria to acknowledge validity of non-Western knowledge [48]<br>Clearly define preferred epistemic frameworks, intended audience and knowledge producer in grant objectives [48]<br>• Invite LMIC researchers to join funding panels and advisory boards [47] |
| **Ensure in-country researchers' direct involvement with funder** | • Create opportunities for local organisations to provide feedback to donors directly, rather than via intermediaries or research partners [25,26]<br>• Global South researchers should insist on clear division and documentation of decision-making power and accountability [51]<br>• Global South researchers are advised to contact funders, including to discuss grant agreements, to ensure their participation [51] |
| **Reimagine risk perception, project management and evaluation** | • Reduce the use of strict timelines and templates [52]<br>• Assess and manage risk to centre accountability to crises-affected communities [55]<br>• Redefine reporting and evaluation metrics to centre the values and visions of communities [55] |
| **Research Lifecycle** | |
| **Collaboratively design contextually specific research methods** | • Meaningfully engage individuals and communities with less power [56]<br>• Design research according to local or regional priorities [48]<br>• Design research proposals collaboratively between non-Western and Western researchers [26] |
| **Collect data in contextually appropriate ways** | • Work closely with local researchers during data collection process [55]<br>• Consider the gendered impacts of data collection methods [56] |
| **Consider the impacts of data analysis methods** | • Engage front-line researchers and study populations in conducting intersectional gender analysis [56] |
| **Engage in equitable research dissemination** | • Engage front-line researchers and study populations in dissemination of findings [56] |
| **Teaching and Curricula** | |
| **Review and reframe course materials** | • Examine social, economic, and political determinants of global health and poverty [46]<br>• Incorporate local and regional perspectives about health into curricula [46]<br>Reflect on dominant and excluded voices and narratives in reading lists [50]<br>• Use dignified and compassionate images that avoid perpetuating stereotypes [50]<br>• Avoid generalising, pathologising, dehumanising, objectifying and minimising language [50]<br>• Include diverse knowledges in learning outcomes and assessments [46,50]<br>• Develop decolonial guidelines for external speakers, including language, image use and content [50]<br>• Invite student feedback and ensure signposting to education-related complaints procedures [46] |

(*Continued*)

**Table 3.** (Continued)

| Area of Recommendation | Suggested actions |
| --- | --- |
| **Create open and inclusive classroom environments** | • Develop awareness of how teacher positionality in terms of race, class, gender, sexuality, and ability status may influence interactions with students and the learning environment [46,50]<br>Teaching staff to focus on student participation and confidence [50]<br>• Teaching staff to disrupt student dominant voices and create space for all voices and opinions [50]<br>• Train teaching staff in decolonial classroom management techniques [50]<br>• Invest resources to recruit diverse graduate students and staff [46,50] |

organisations should commit to recruitment policies and strategies that increase staff diversity in Global North offices. Peace Direct [26] suggests using the *Women of Color Advancing Peace, Security, and Conflict Transformation*'s "Orgs in Solidarity" 12-point solidarity statement [57] as strategic point of reference.

A fourth step is to reframe internal and external communication policies away from the predominant Eurocentric, benevolent, white saviourist view of aid [23,53]. This process starts by advancing an understanding of the harm caused by continuing to use images of people of colour as dependent and incompetent [23]. Kertman [54] recommends centring experience and prioritising agency, as opposed to pain and suffering; mainstreaming race similar to gender, ability, and class; and creating equitable storytelling to influence public perceptions of global poverty. Terms with racist, supremacist and paternalistic connotations should be removed from internal and external communications. For example, "beneficiaries" removes agency and power from communities who are often the first and most active responders to crises. "Aid" infers that humanitarian action and funding is based on charity, without recognising colonial histories of exploitation. Some argue that to redress inequitable dynamics, aid should be reframed as reparations [26,55]. Pivoting towards a racial justice oriented, historically accountable, and reparations-based framing of aid would reduce the need for such organisations to fundraise using current frames that rely on guilt, pity, and White Saviourism [54].

Finally, humanitarian actors need to broaden their allyship and political engagement to move away from notions of "neutral" and "apolitical" humanitarian aid [53]. Aloudat and Khan's analysis [53] shows how decolonisation frameworks in the humanitarian sector have so far sidestepped critiques of monopoly, misuse, or abuse of power. They suggest the humanitarian community's response to the attack on Gaza in May 2021 is an example of the problematic depoliticisation of humanitarian aid, where the humanitarian community stood largely silent, or gave muted calls to "stop attacks", thereby creating a false equivalence of guilt between the Palestinians in Gaza and Israeli military power [53]. Instead, they suggest actors should locate their humanitarian operations within the political arena(s), and link aid with other social justice issues such as anti-racism and the climate crisis [53].

**Research partnerships and conceptualisation.** The second area of guidance we identified relates to research partnerships and research conceptualisation. A key inequitable research practice is the tendency for those in the Global North to design interventions or research from the Global North with little to no coordination or engagement with the project participants closest to the setting/issue [47]. Internalising and integrating local knowledge and solutions [47] would require White, Northern actors to "recognise, prioritise and mainstream the knowledge and views of the world constructed outside the frame and experience of Whiteness,

and to stand back to allow others to speak for themselves" A key inequitable research practice is the tendency for those in the Global North to design interventions or research from the Global North with little to no coordination or engagement with the subject people of these with the project participants closest to the setting/issue [47]. Internalising and integrating local knowledge and solutions [47] would require White, Northern actors to "recognise, prioritise and mainstream the knowledge and views of the world constructed outside the frame and experience of Whiteness, and to stand back to allow others to speak for themselves" [23]. Accordingly, Khan et al. [47] contend that international staff should have sufficient local intelligence, and, if suitable, should be employed at the level of technical advisers or coordinators, rather than decision-makers [47]. International staff should not be assumed to be "experts", and their employment should not supersede non-Western and local expertise and capacity. Caution should be exercised, however, against assuming international staff to be the natural "experts"; their employment should not supersede non-Western and local expertise and capacity.

Guidance critiques the common concept of "capacity building", which reinforces discriminatory and racist perceptions of non-white populations, suggesting that local communities and organisations lack capacity [26]. Lack of capacity and/or expertise should not be assumed, and HIC researchers should instead focus on connecting colleagues to contextually-specific and locally-prescribed resources they require to undertake research or implement interventions [52] This is understood by some as "capacity strengthening" or "capacity bridging" [52]. There is also a need to interrogate how capacity is being evaluated and by whom. If, for example, education qualifications from Western institutions are regarded more highly than those from non-Western institutions, degree valuation needs to be re-envisioned at the institutional level and sector-wide, rather than connecting LMIC colleagues to more Western training [48].

With regards to South-North research partnerships, Rasheed [51] recommends that Global South lead researchers ensure that their whole team, especially those working with participants directly, agree the project strategy from the outset. Clear guidelines should be established to ensure that Global South lead researchers are copied into any communications between global partners, and the rest of the Global South research team. [51]. When such equitable partnerships are sought, Rasheed [51] warns that complaints and escalations targeting Global South researchers' technical skills should be anticipated as a counter strategy. Documenting all project decisions is therefore key.

Various recommendations were made in relation to establishing and evaluating long-term collaborations. Evaluations can increase mutual accountability and improve relationships and could be achieved through annual partnership reviews or otherwise inviting feedback from partners [52]. Little other practical guidance on partnership evaluation mechanisms or best practice were identified in this body of literature. Guidance suggests that strict timelines and templates for project management should be reduced [52]. Aid Re-imagined [52] recommend centring metrics focused on community values, such as "Outcome Harvesting". The suggestion to stop using measures for evaluation altogether was also mentioned [52] but examples of this practice were not identified In addition to recalibrating North-South partnerships, there are calls to encourage and implement South-South collaboration. Abimbola et al. [45], for example, call for the need to "build Southern networks to affirm our ownership of global health".

**Funding for research and projects.** The third area of guidance identified in this review relates to funding for research and interventions. Global health research funding is affected by unequal, colonial power dynamics that prioritise outsider perspectives over local needs [48]. Localising funding was mentioned as a key step towards decolonising global/humanitarian research and practice, [45]. Funding local community and civil society actors needs to be

considered when setting funding criteria and measuring intervention success, as it contributes towards epistemic justice by placing value on knowledge created in places of intervention. Consequently, Koum Besson [48] regards localising 'Call for Proposals' and review processes as important levers in decolonising global health research funding. When drafting funding calls and evaluation criteria, local communities and researchers should be both the intended audience, and producers of knowledge [48]. Additionally, funders should provide clear reasons for why their calls require dominant Euro-North American-centric tools, or otherwise allow Global South researchers either to apply interpretative frameworks and/or develop new methodological tools that arise out of local epistemic contexts and/or lived experiences [48]. Epistemic justice can also be achieved by creating spaces for researchers from Global South institutions on funding panels and advisory boards that set research agendas [47]. Beyond recalibrating funding criteria, donors must provide opportunities for local organisations to share feedback directly, rather than via intermediary partners [25]. Donors should listen to concerns about power imbalances in the funding system, and specifically provide opportunities for critiques of their own practices [26].

Rasheed [51] recommends researchers located in low-income countries be particularly vigilant during the grant agreement process. Before signing agreements, researchers should insist on being part of communications with the funder, meet with their own administrative and legal teams to examine contracts and, if comfortable to do so, discuss concerns with the donor. LMIC researchers should be understand accountabilities related to whether a grant is a primary or sub-award, and who they will ultimately be reporting to. Rasheed [51] suggests low-income country researchers to clarify and document the project's decision-making processes. [51].

One major obstacle to supporting locally led humanitarian action is prevailing approaches to risk management. Due diligence and risk analysis instruments present entry and funding barriers for smaller, local and grassroots organisations, impose Global North standards, and define deficits from this vantage point [55]. Funding needs to be made more accessible and inclusive for local groups and communities. While some call for the need to "trust generously" and "fund courageously' [26,52], others describe the need to "reimagine risk", through the use of new approaches and tools which locate accountability within crises-affected communities, rather than Northern donors [55].

**The research lifecycle.** The fourth area of guidance relates to the research process itself, including design, data collection and analysis, as well as dissemination and authorship practices. Research teams should develop plans for the meaningful participation of individuals and communities with less power [56]. Global South practitioners who are invited to participate in research are usually relegated to the role of "expert in the field", with Global North researchers leading the research proposal and design, methodology, tools and data collection processes [26]. Global South researchers are positioned as "field assistants" and, once the data collection is completed, Global North researchers take the lead in producing outputs and are credited with the work [26]. Peer research is a participatory research method in which people with lived experience of the issues being studied take part in directing and conducting the research. According to Peace Direct, it has the potential to give communities "the power to communicate their own priorities rather than [those] being left to well-meaning philanthropists in the West" [26].

As Koum Besson [48] notes, research design often hinges on grant objectives. These are often based on "addressing gaps in the literature" or the generation of generalisable findings. When the need to produce knowledge is based on what is globally known, rather than locally known or not known, local knowledge is automatically positioned as less credible. Research solely based on what is known globally can clash with Global South researchers' approaches to

making sense of and addressing social structures that disadvantage communities in their context. As such, contextually-specific research designs should be prioritised, that consider the political, social, economic, and historical contexts and power hierarchies of the research setting [56]. In line with this, research rationales should align with national or regional research priorities (e.g., Africa Centres for Disease Control and Prevention, national public health institutes, local research centres, etc.) over international agendas [48]. Recognising who is driving the need for the study is key in efforts to decolonise the humanitarian research process [48].

Data collection should be undertaken in collaboration with local researchers, to ensure cultural and contextual appropriateness. Research participants may feel more comfortable expressing their perspectives in their own languages and with people they feel they can trust [55]. Follow-up community surveys can be conducted to assess how people feel they were treated during data collection [55]. Following COVID-19-related movement restrictions, remote data collection has become common practice, but poses challenges around gender, racial and other inequities. For instance, if women have less access to the internet, mobile phones and digital literacy, they may be less able to participate in research [56].

Researchers should also consider the impacts that their data analysis methods have on communities. Feminist approaches such as those discussed in Singh et al. [56] advocate for employing intersectional analysis to centre participants' voices and knowledges, and uncover any colonial gender, racial and other power hierarchies that may be embedded within the research process itself. This approach forms part of an "ethics of care" approach that values study participants more than the data they generate and attempts to redress the traditionally unequal power dynamics between ostensible "objects" and "subjects" of research [56].

Humanitarian research is commonly conducted in a host country, while results are disseminated in the language of Principal Investigators, who are disproportionately located in the Global North [45,48,56]. The question subsequently evolves around who benefits from the data and knowledge generated, and whether, and if so, how, participants experience benefits from the study Frontline researchers and study participants should be central to disseminating findings, which in turn should be used to challenge unjust systems and policies to deliver transformative and equitable programmes [56]. As such, knowledge generated should aim to meet local, national or regional needs first, and manuscript preparation after [48]. The focus should be on how the evidence is being used, where it is stored and who it is helping, rather than publication. When articles are published, publication in Global South journals should be encouraged [48].

**Teaching and curricula.** The fifth area of guidance relates to teaching and curricula. At LSHTM, a workstream group called "Decolonising the Curriculum" has worked together with the Centre for Excellence in Learning and Teaching to develop a "Decolonising the Curriculum" toolkit [50]. This internal toolkit provides "guidance, links to other resources, and examples of how staff have brought decolonial perspectives into their teaching practice" and is accessible for all staff and students [50]. Similarly, the FAIR Student Toolkit [46] was developed to facilitate student engagement with questions around race, racism, (de)coloniality, and anti-racism in global health. The key recommendation from these two toolkits relate to inclusive classroom environments and reviewing and reframing course materials.

A key recommendation of these toolkits is the creation of open and inclusive classroom environments that provide opportunities for student-led inquiry and feedback [46,50]. Increasing student participation and confidence depends on the educator proactively disrupting voices of dominant students and in doing so, creating space for all voices and opinions [50]. Participation can also be improved by training teaching staff classroom management skills through a decolonising lens, as well as asking students open questions to facilitate their contribution to knowledge production by encouraging them to bring their values and

experiences to the classroom [50]. Awareness of how the lecture or seminar leader's' position-alities in terms of race, class, gender, sexuality, and ability status influence interactions within the learning environment is also required [46,50]. While these tasks are the responsibility of lecturers and seminar leaders, organisational leadership also plays a key role in this endeavour. Higher education institutions should invest resources to create supportive environments and recruit non-white staff to address lack of diversity in the teaching body, as well as provide financial resources towards institutional Equality Diversity and Inclusion (EDI) committees [46,50,55].

The second key recommendation identified in these toolkits is to review and reframe course materials from a decolonial perspective. This includes the exploration of alternative (local and regional) knowledge sources that incorporate these contexts to de-centre Western knowledge and knowledge production [46]. A reading list review should critically reflect on dominant voices and narratives, identify excluded voices and narratives, examine the gender and ethnic-ity balance of the reading list, as well as geographic coverage, place of publication and language of texts, and ask what kinds of sources are presented to be of academic value and why [46]. Module convenors, lecturers and seminar leaders should carefully consider using dignified and compassionate images that avoid perpetuating stereotypes of who is the receiver of aid and assistance, amongst others [cf.23,26,50]. Close attention should be paid to how language is used, avoiding generalising, pathologising, dehumanising, objectifying and minimising termi-nology [23,26,50,54]. Modules can also update learning outcomes and assessments to encom-pass knowledge produced in the countries/regions that are considered by the teaching activities [50]. Learning objectives should be built on recognising multiple forms of knowledge and include the perspectives of study participants. Materials and reading lists should be updated with critical literature on reformative and transformative viewpoints and equitable approaches to improving health worldwide [46,50].

## Discussion

This is the first review of its kind to synthesise key recommendations and guidance from both academic and grey literature on decolonising global/humanitarian health research, partner-ships, teaching, organisational structures and other practices. It does not reject humanitarian-ism altogether; instead, it presents a concrete set of guidelines to support humanitarian actors in decolonising their work. Our principal finding is that global/humanitarian health actors can decolonise their work by decentralising power, redistributing resources, critically reflecting on their work in the context of the broader socio-political landscape and recovering and centring perspectives from those in the Global South, which have historically been side-lined in global/humanitarian health discourse and practice. Specifically, the review identified five key themes surrounding the practical guidance on decolonising humanitarian health research, teaching and practice: organisational structure, strategy and engagement; research partnerships and conceptualisations; funding for research and projects; the research lifecycle; and teaching and the curriculum. We also identified some pitfalls and several important gaps in the reviewed body of literature with regards to authorship, awareness and engagement with broader socio-political questions, details around implementation of research, teaching and partnerships, as well as race as an analytical category. Only 15 documents contained practical guidance, which points to the scarcity of practical advice in the available literature and the importance of our contribution.

The aim of this review was to focus on practical guidance on decolonising. In addition to more abstract recommendations, it has identified a set of tangible and practical steps that researchers, educators and practitioners can take, albeit mainly aimed at those in the Global

North. Examples of such steps include removing commonly used development language, such as "beneficiaries" or "aid", as well as implementing approaches like "peer research". Several examples of where guidance has been implemented successfully were also given, including resource reallocation through a so-called "networked model" recently employed by the NGO IPAS, or seed funding for partnership meetings before proposal submissions, provided by funders at ELHRA/R2HC.

In the wider literature, there have been strong warnings of the potential pitfalls of attempting to decolonise a field of practice or institution [58]. It is worth noting that, in this review, we identified recommendations on decolonising which do not fully decouple humanitarian practices from colonial power relations and logics. For example, some authors propose the need to "trust generously" and "fund courageously" under the theme of funding of research and projects [26,52]. While the aim of this recommendation is to make funding more accessible, positioning the funding of those in the Global South and/or of community groups as requiring more "trust" and "courage" from a funder, and thus as inherently financially risky, fails to question why those in the Global North hold and control research funds, and feeds into a paternalistic and colonial discourse characteristic of global/humanitarian health. Others avoid this terminology, recommending that risk ought instead to be "re-imagined" [55] or to ensure that Global South researchers can shape research funding priorities and decisions via funding panel and advisory board membership [48]. Where contradictory guidance was identified in this literature review, we discussed which recommendations more closely aligned with decolonial theory and selected the most appropriate language to feed into the final key guidance provided in Table 3.

Several gaps were identified in the literature on decolonising global/humanitarian health. The fact that this scoping review only identified 15 relevant documents highlights the need for more practical guidance on decolonising aid. That the majority of papers we reviewed in full concerned the humanitarian sector also reflects the decades-long debates and discussion around reparations, decolonisation and humanitarian assistance. More significant, however, is that only one paper [56] spoke to decolonising research in humanitarian settings, which points towards future avenues for enquiry.

One of the major gaps in the included literature is guidance written by, and for, researchers based in the Global South–only one paper fell into this category. Rasheed's paper [51] was written by and for researchers from low-income countries who enter into academic research partnerships with individuals and institutions in high-income countries. It is noteworthy that, while there was much overlap in guidance provided by the documents identified in this review, this paper has a different focus and offers unique recommendations in comparison to the other documents. Rasheed's [51] arguments are underpinned by post- and decolonial thinkers such as Franz Fanon [59] and focus on navigating what the author considers to be an inherently violent process; namely, when academic researchers in the Global South seek to challenge power asymmetries on their own terms. Consequently, Rasheed's guidance has a unique focus on putting concrete logistical measures in place in anticipation of potentially marginalising reactions from partners to these challenges. We suggest that guidance on decolonising which emanates from the Global South is not likely to be the same as that written in the Global North.

The lack of guidance written specifically by and/or for those in the Global South also raises important questions around the potential misappropriation of decolonisation in the institutional setting [58] as well as why and for whom we are decolonising global health [43]. Still, the potential for misusing the decolonisation agenda does not mean we should do away with decolonising, which is a rich tradition of scholarship and practice itself emanating directly from the Global South and which is responsible for what we know about how to challenge colonial

patterns of domination [60,61]. Instead, while continuing to reflect on critical questions around the use and misuse of the decolonisation agenda, we recognise that the structural constraints and barriers identified in this review may account for the lack of identified guidance produced by those based in the Global South.

Another gap we have identified in the literature on decolonising humanitarian health and practice is the absence of race as a central analytical category. Though power is analysed and centred throughout these documents, race as a specific social category and power structure is not centred in the majority of the documents. Interestingly, although race forms a central part of the historical narrative of colonialism described in their background sections, almost none of the documents make specific recommendations for practical action regarding race. Though reasons for this can vary, we note that race is generally not mainstreamed in organisational communications or teaching and research practice in the way that gender or class have been. The 2020 global Black Lives Matter protests in that sense filled a void in public discourse around the real and lethal effects of racism. One of the key recommendations regarding organisational structure, strategy and engagement that have been made in a post-2020 paper that we reviewed is to mainstream race when reframing communications [54]. This can be done, for example, using an explicit racial justice lens when developing communications policies. A notable example of a reviewed document that does centre race throughout their analysis is Peace Direct [26]. They regard country offices as an output of structural racism in the humanitarian sector. They also comment on the relationship between race and power in the context of procedural challenges that stem from systemic racism and financial, institutional and epistemic power centralisation in the Global North [26]. This is a crucial finding of this literature, and we invite future scholarship and practical guidance to reflect on practical steps towards racial justice in humanitarian health research, teaching, and practice.

Two further areas for which less practical advice was identified were calls to broaden the allyship and political engagement of institutions and organisations, and to encourage and implement South-South collaboration. The former relates to how organisations and institutions should, if attempting to decolonise their work, challenge the status quo of the global political economy and connected global governance structures. The recommendation identified in this area is for humanitarian actors to expressly locate their humanitarian operations within the political arena(s), and link aid with other social justice issues [53]. The rejection of aid as 'apolitical' and 'neutral' is certainly a key part of engaging in its decolonisation, but questions arise as to what this might look in practice–especially given how existing humanitarian principles emphasise neutrality [62]. This review also identified calls to encourage and implement South-South cooperation in order to build Southern Networks and increase the Global South's stake in, and ownership of, global health structures [45]. This should be done while avoiding what Abimbola and colleagues call "elite capture" whereby only more privileged Global South actors access benefits [45]. While there is a rich body of literature on the history of South-South Cooperation [63–65], further guidance on how this can or has been implemented in global/humanitarian health was not identified in this review but would be interesting for further research.

This scoping review has some limitations. Firstly, we focussed the review on documents that contained practical guidance. Therefore, much of the initially identified literature, which provided diverse and unique understandings of decolonising global/humanitarian health, had to be excluded as it was not practical in nature. Narrowing down our criteria to focus on practical guidance alone enabled our review to present clear and pragmatic guidance for researchers, educators and practitioners in global/humanitarian health on how they might decolonise their work. Providing this concrete set of actions is a key strength of this review.

Furthermore, due to the large number of records generated by the broad search terms used for the database searches, we decided to focus only on documents that explicitly reference decolonising. The use of this inclusion criterion presents some inherent limitations, given that there are a variety of terms either implicitly or explicitly associated with decolonisation. There are, for example, papers that do not explicitly use the terms "decolonisation" or "decolonising", but speak about "localisation" [66], "equitable partnerships" [67,68], "Black Lives Matter" [69] or "indigenisation" [70], all of which could be considered aspects of decolonising. This meant therefore that we left out some potentially useful guidelines from our main findings. For example, guidance on ensuring long-term research collaborations and their evaluation in the Lancet Palestinian Health Alliance [71] shed light on structural imbalances perpetuating inequitable research and interventions but did not explicitly refer to decolonising, so was excluded. We also excluded guidance that focused solely on localising global and humanitarian health [66] equitable authorship and dissemination, [72], and upstream approaches to global health training [73]. By only including sources that use the terminology of "decolonising", we have highlighted some inherent difficulties with the practical application of this agenda given that, not unexpectedly, this process is understood differently across the global/humanitarian health literature.

Some methodological limitations include that this review only included English-language documents. A multi-lingual review might yield much richer results, including those written by and for Global South researchers, of which we only found one paper in English. In addition, reducing the geographic scope of the review, may also have diminished its conceptual, narrative and practical richness while reproducing colonial academic practices that narrow the dissemination of results to certain actors and populations [43]. Finally, scoping reviews such as ours contain some inherent limitations, chiefly around enabling a narrative and descriptive appraisal of the literature, as well as difficulties in synthesising a wide variety of sources (blogposts, peer-reviewed papers, Moodle courses etc.) [38].

The key strength of this review is its synthesis of concrete actions that researchers, educators, and practitioners of global/humanitarian health can take to decolonise their work. At the time of writing, practical guidelines on how to decolonise work in this field had neither been reviewed, nor collated or synthesised. As such, we make a unique contribution to efforts to decolonise the humanitarian and global health sector and hope that this review will be helpful in steering and guiding concrete actions by practitioners, researchers and educators. Furthermore, this review is timely and relevant given that many of the conversations on decolonisation that were catalysed by the global Black Lives Matter protests in 2020 have resulted in various grey and peer-reviewed publications. We started this review in 2022, which was an opportune moment to review and synthesise these guidelines. Our synthesis therefore comes at a moment in which many organisations, researchers and educators show a pronounced interest in decolonising their work and would benefit, we hope, from a distilled set of practical guidelines and concrete actions they can take.

## Conclusion

This review provides the first known attempt to map and synthesise the practical guidance on decolonising humanitarian/global health. We present concrete guidelines to support humanitarian actors in decolonising their work, identifying five key themes: organisational structure, strategy and engagement; research partnerships and conceptualisations; funding for research and projects; the research lifecycle; and teaching and the curriculum. Our principal finding is that global/humanitarian health actors can decolonise their work by decentralising power, redistributing resources, critically reflecting on their work in the context of the broader socio-

political landscape and recovering and centring perspectives from those in the Global South. As well as this concrete guidance, we identified important potential pitfalls and areas for future research that may act as a warning to those seeking to embark on a journey of decolonising their work.

We suggest that further research is required to more clearly illuminate the path towards decolonisation by humanitarian/global health actors. There is an overall lack of guidance which is practical, centred race as an analytical category, engages with the importance of the wider socio-political environment and details how South-South cooperation can be implemented. It is also clear that there is a need to clarify the relationship between decolonisation and related concepts as there is evidently some conflation of different terms. Finally, there is lack of identified guidance produced by those based in the Global South, arising from the structural constraints and barriers identified in this review. We call for humanitarian researchers and actors to (continue to) build on the growing literature and critical momentum around decolonising authorship, dissemination, funding etc., to dismantle those barriers so that Global South researchers will increasingly lead development of practical recommendations on decolonising global/humanitarian health from their perspective.

## Supporting information

**S1 Appendix. Database search terms.**
(DOCX)

## Author Contributions

**Conceptualization:** Amber Clarke, Katharina Richter, Michelle Lokot, Althea-Maria Rivas, Sali Hafez, Neha S. Singh.

**Data curation:** Amber Clarke, Michelle Lokot.

**Formal analysis:** Amber Clarke, Katharina Richter.

**Funding acquisition:** Amber Clarke.

**Investigation:** Amber Clarke, Katharina Richter.

**Methodology:** Amber Clarke, Katharina Richter, Michelle Lokot.

**Project administration:** Amber Clarke, Katharina Richter.

**Resources:** Amber Clarke.

**Supervision:** Amber Clarke, Katharina Richter, Michelle Lokot.

**Writing – original draft:** Amber Clarke, Katharina Richter.

**Writing – review & editing:** Amber Clarke, Katharina Richter, Michelle Lokot, Althea-Maria Rivas, Sali Hafez, Neha S. Singh.

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
