## [Decision Letter · Decision Letter 0]

29 May 2024

PGPH-D-24-00057

Decolonising humanitarian health: A scoping review of practical guidance

Dear authors,

Thank you for submitting your manuscript to PLOS Global Public Health. After careful consideration, we feel that it has merit but does not fully meet PLOS Global Public Health’s publication criteria as it currently stands. Therefore, we invite you to submit a revised version of the manuscript that addresses the points raised during the review process.

We look forward to receiving your revised manuscript.

Kind regards,

Andreas K Demetriades, MBBChir, MPhil, FRCSEd, FEBNS.

Academic Editor

Journal Requirements:

Additional Editor Comments (if provided):

Encouraging comments from the peer review process

Please consider these when revising and resubmit within a month

Reviewers' comments:

Reviewer's Responses to Questions

**Comments to the Author**

1. Does this manuscript meet PLOS Global Public Health’s publication criteria? Is the manuscript technically sound, and do the data support the conclusions? The manuscript must describe methodologically and ethically rigorous research with conclusions that are appropriately drawn based on the data presented.

Reviewer #1: Yes

Reviewer #2: Yes

2. Has the statistical analysis been performed appropriately and rigorously?

Reviewer #1: N/A

Reviewer #2: N/A

3. Have the authors made all data underlying the findings in their manuscript fully available (please refer to the Data Availability Statement at the start of the manuscript PDF file)?

Reviewer #1: Yes

Reviewer #2: Yes

4. Is the manuscript presented in an intelligible fashion and written in standard English?

Reviewer #1: Yes

Reviewer #2: Yes

5. Review Comments to the Author

Reviewer #1: Clarke et al. present their work on decolonising humanitarian health in the Global South. The manuscript has merrit, the topic is truly relevant and reflects an ongoing issue encountered in humanitarian aid. The recommendations are stated in a comprehensive and clear way, and the data has been selected accordingly.

Overall, the mansucript is very long (37 pages) and some chapters are repetitive. A shortening of the introduction and the written results of the five key themes would possibly encourage more readers to stay focussed and continue reading. The manuscript has aspects of a philosophical essay and should remain informative but neutral.

Furthermore, a paragraph on detailing the cultural background of the authors seems out of context.

Reviewer #2: The term global health was born parallel to the expansion of colonialism and racism as a tool to fight against the inequalities provoked by them. This sounds like a bizarre paradox. When talking about humanitarian health, racism surfaces all levels of healthcare, from the clinical practice to the most sophisticated investments and so-called non-profit programs. Therefore, it sounds obvious that any action in this field should come behind a willingness to fight racism and its colonial heritage by renouncing to the power wielded by its main agencies.

The authors introduce their review through an intense and extremely clear text, where the importance of terminology is highlighted, as well as the intrinsic problem of decolonization in general terms, and its inherent linkage to racism. The introduction is extensive, well organized, touches the most relevant gaps in the field, and finalizes by ensuring the real purpose of their review.

The methods are robust and well dissected. The authors access to the most relevant not only scientific sources, but also other familiar networks, increased the quality of their search, reaching a much realistic view of the problem. There was a massive data extraction with the goal of clearly understanding the real situation.

The idea of introducing the difficulties and limitations of the employed terminology in different reviewed studies when describing the methodology is clearly not common, but in my view it helps understanding the results display from a better perspective.

The authors explain nicely how and when they use the terms High-Income Country (HIC) and Low/Middle Income Country (LMIC) vs. Global North and Global South. However, it might be helpful to highlight the source and date they used to classify HIC and LMIC.

Regarding the structure and strategy, the authors have summarized the efforts for decolonization into 5 specific actions which sound absolutely necessary. A key step to start ‘decolonizing anything’ seems to be including ‘local actors’ into the action. Therefore, their third suggestion should be a must in every program. This is also the main core of the research action, and I fully agree with the authors at this point.

Running a step forward into funding research and academics, the main problem appears when realizing that ‘Global North’ decides where to inject funding, and how to take actions. This should be obviously guided by local needs, so, again local actors are needed as key. The toolkits the authors are providing sound like interesting materials to be explored.

The discussion flies smoothly through all the identified points to action regarding decolonization. It touches three relevant aspects that should be really considered as key: race, south-south collaboration, and politics.

It is worth to mention the tremendous exercise of self-criticism that we must make from the Global North before giving any attempt to work in the field.

Therefore, I would like to THANK and congratulate the authors for such an outstanding review.

6. PLOS authors have the option to publish the peer review history of their article (what does this mean?). If published, this will include your full peer review and any attached files.

**Do you want your identity to be public for this peer review?** For information about this choice, including consent withdrawal, please see our Privacy Policy.

Reviewer #1: **Yes: **Vicki Marie Butenschoen

Reviewer #2: **Yes: **Pablo González-López

---

## [Decision Letter · Decision Letter 1]

20 Aug 2024

Decolonising humanitarian health: A scoping review of practical guidance

PGPH-D-24-00057R1

Dear authors

We are pleased to inform you that your manuscript 'Decolonising humanitarian health: A scoping review of practical guidance' has been provisionally accepted for publication in PLOS Global Public Health.

Best regards,

Andreas K Demetriades, MBBChir, MPhil, FRCSEd, FEBNS.

Academic Editor

Thank you for your submission to this journal.

The peer review has been very positive after the revised manuscript has been assessed and this is now recommended for publication.

Reviewer Comments (if any, and for reference):

Reviewer's Responses to Questions

**Comments to the Author**

1. If the authors have adequately addressed your comments raised in a previous round of review and you feel that this manuscript is now acceptable for publication, you may indicate that here to bypass the “Comments to the Author” section, enter your conflict of interest statement in the “Confidential to Editor” section, and submit your "Accept" recommendation.

Reviewer #1: All comments have been addressed

Reviewer #2: All comments have been addressed

2. Does this manuscript meet PLOS Global Public Health’s publication criteria? Is the manuscript technically sound, and do the data support the conclusions? The manuscript must describe methodologically and ethically rigorous research with conclusions that are appropriately drawn based on the data presented.

Reviewer #1: Yes

Reviewer #2: Yes

3. Has the statistical analysis been performed appropriately and rigorously?

Reviewer #1: Yes

Reviewer #2: N/A

4. Have the authors made all data underlying the findings in their manuscript fully available (please refer to the Data Availability Statement at the start of the manuscript PDF file)?

Reviewer #1: Yes

Reviewer #2: Yes

5. Is the manuscript presented in an intelligible fashion and written in standard English?

Reviewer #1: Yes

Reviewer #2: Yes

6. Review Comments to the Author

Reviewer #1: The authors responded to all my comments, and in one case discussed why they want to keep the authors personal biography in the manuscripts-which sounded reasonible and therefore can be accepted. Congratulations on their extensive effort!

Reviewer #2: I feel that the authors agreed and included most of our recommendations in a right manner.

7. PLOS authors have the option to publish the peer review history of their article (what does this mean?). If published, this will include your full peer review and any attached files.

**Do you want your identity to be public for this peer review?** For information about this choice, including consent withdrawal, please see our Privacy Policy.

Reviewer #1: **Yes: **Vicki Marie Butenschoen

Reviewer #2: **Yes: **Pablo González-López
